# Mechanistic Insights into the Link between Gut Dysbiosis and Major Depression: An Extensive Review

**DOI:** 10.3390/cells11081362

**Published:** 2022-04-16

**Authors:** Sharma Sonali, Bipul Ray, Hediyal Ahmed Tousif, Annan Gopinath Rathipriya, Tuladhar Sunanda, Arehally M. Mahalakshmi, Wiramon Rungratanawanich, Musthafa Mohamed Essa, M. Walid Qoronfleh, Saravana Babu Chidambaram, Byoung-Joon Song

**Affiliations:** 1Department of Pharmacology, JSS College of Pharmacy, JSS Academy of Higher Education & Research, Mysuru 570015, Karnataka, India; sonalisharma578@gmail.com (S.S.); bray365@gmail.com (B.R.); tousif.a.h7@gmail.com (H.A.T.); tuladharsunanda4@gmail.com (T.S.); ammahalakshmi@jssuni.edu.in (A.M.M.); 2Centre for Experimental Pharmacology and Toxicology, Central Animal Facility, JSS Academy of Higher Education & Research, Mysuru 570015, Karnataka, India; 3Food and Brain Research Foundation, Chennai 600094, Tamil Nadu, India; agrathipriya@gmail.com; 4Section of Molecular Pharmacology and Toxicology, Laboratory of Membrane Biochemistry and Biophysics, National Institute on Alcohol Abuse and Alcoholism, National Institutes of Health, Rockville, MD 20892, USA; wiramon.rungratanawanich@nih.gov; 5Department of Food Science and Nutrition, CAMS, Sultan Qaboos University, Muscat 123, Oman; drmdessa@gmail.com; 6Aging and Dementia Research Group, Sultan Qaboos University, Muscat 123, Oman; 7Q3CG Research Institute (QRI), Research and Policy Division, 7227 Rachel Drive, Ypsilant, MI 48917, USA; walidq@yahoo.com

**Keywords:** depression, gut microbiota, gut dysbiosis, gut–brain axis, short-chain fatty acids, serotonin, hypothalamus–pituitary–adrenal axis, major depressive disorder

## Abstract

Depression is a highly common mental disorder, which is often multifactorial with sex, genetic, environmental, and/or psychological causes. Recent advancements in biomedical research have demonstrated a clear correlation between gut dysbiosis (GD) or gut microbial dysbiosis and the development of anxiety or depressive behaviors. The gut microbiome communicates with the brain through the neural, immune, and metabolic pathways, either directly (via vagal nerves) or indirectly (via gut- and microbial-derived metabolites as well as gut hormones and endocrine peptides, including peptide YY, pancreatic polypeptide, neuropeptide Y, cholecystokinin, corticotropin-releasing factor, glucagon-like peptide, oxytocin, and ghrelin). Maintaining healthy gut microbiota (GM) is now being recognized as important for brain health through the use of probiotics, prebiotics, synbiotics, fecal microbial transplantation (FMT), etc. A few approaches exert antidepressant effects via restoring GM and hypothalamus–pituitary–adrenal (HPA) axis functions. In this review, we have summarized the etiopathogenic link between gut dysbiosis and depression with preclinical and clinical evidence. In addition, we have collated information on the recent therapies and supplements, such as probiotics, prebiotics, short-chain fatty acids, and vitamin B12, omega-3 fatty acids, etc., which target the gut–brain axis (GBA) for the effective management of depressive behavior and anxiety.

## 1. Introduction

Depression is defined as a serious mental illness characterized by depressed mood and long-lasting anhedonia in normal life activities for at least 15 days (American Psychiatric Association, 2013). According to the World Health Organization, depression causes major mental illness, affecting more than 300 million people globally (2017–2020). Depression, especially major depressive disorder (MDD), is the second leading cause of disability and the most commonly affective disorder diagnosed in millions of adolescents and adults worldwide within the age bracket of 15–44 years old [1]. The main physical symptoms include insomnia or hypersomnia, fatigue, tiredness, disturbed appetite, social isolation, unintentional weight change, agitation or restlessness, fidgeting, crying, back pain, or headaches [2]. The prevalence is more common in women than men [3]. The main psychological symptoms include frustration, irritation, negative thinking patterns, mood swings, poor concentration, difficulty in making decisions, focusing, or thinking, or feelings of low self-worth, guilt or hopelessness, and in severe cases, thoughts of suicide or death [4,5,6]. A variety of factors such as genetic, environmental, psychological, and metabolic causes account for the etiology of depression. The common age range for the onset usually falls within 15–30 years of age [7].

### 1.1. Depression and Gut Microbiota

Recent studies in the human microbial ecosystem have shown the important physiological role of gut microbiota (GM) in maintaining the gastrointestinal (GI) tract and hormonal, immune, and neural homeostasis [8]. Gut dysbiosis (GD) is defined as an abnormal alteration in the composition and abundance of GM affecting the host’s homeostasis [9]. Several clinical and pre-clinical studies have reported a causal link between depression and GD [10,11,12,13], which alters brain activities via GBA [14,15]. The neural transmission (both the hypothalamus–pituitary–adrenal (HPA) axis and afferent fibers of the vagal nerve) was reported to be disrupted by GD associated with gut leakiness and local inflammation [16], which are in turn connected to anxiety and depression [17,18]. O’Mahony et al. (2009) suggested that anxiety causes an HPA axis imbalance, leading to systemic immune responses and GD [19]. GD aggravates anxiety or depressive behavior and also causes cognitive impairment [2], while maintaining healthy gut by probiotics can improve the anxiety and depression in experimental mice [17].

### 1.2. The Pathological Role of Gut Dysbiosis in Depression

Common triggers for depression are psychosocial problems, sleep disorders, and chronic stress or anxiety, which activate the immune system via the HPA axis. The hyperactivation of the HPA causes increased production of reactive oxygen and nitrogen species that can damage cellular proteins, nucleic acids, and lipids, and decreased levels of antioxidants, such as reduced glutathione and amino acids such as tryptophan and tyrosine [20]. Maes et al. (2008) reported that an increased translocation of a bacterial endotoxin lipopolysaccharide (LPS) from Gram-negative enterobacteria activates the inflammatory response system (IRS), which leads to the elevation of pro-inflammatory cytokines, such as IL-6, IL-1β, and tumor necrosis factor-α (TNFα), and mitogen-induced lymphocyte responses. Similarly, an increased indoleamine 2,3-dioxygenase activity depletes serotonin (5-hydroxytryptamine, 5-HT) levels in the brain, causing psychomotor retardation, malaise, and loss of interest, ultimately resulting in MDD [21]. Targeting the GBA with a probiotic *Lactobacillus rhamnosus* (JB-1) reduced γ-aminobutyric acid (GABA) Aα2 mRNA expression in the prefrontal cortex and amygdala, and thereby decreased corticosterone levels in plasma [22]. In another study, administration of B-immuno galacto-oligosaccharide and fructo-oligosaccharides promoted the growth of beneficial bacteria such as *Bifidobacterium longum*, but reduced the abundance of pathobionts, suggesting an improvement of GM. Supplementation with Lactobacillus strains also decreased the stress-induced HPA activation and corticosterone, while in turn it elevated the levels of brain-derived neurotrophic factor (BDNF) in the hippocampus and 5-HT in the prefrontal and frontal cortex in mice, to protect against the anxiety- and depressive-like behavior [23].

### 1.3. Objectives

In the present review manuscript, we summarized the evidence that the gut dysbiosis is associated with depression, based on preclinical and clinical reports. We also reviewed in a narrative manner the involvement of key pathways, such as the HPA axis activation, serotonin–tryptophan metabolism, neuroinflammation, oxidative stress, and GD, in depression. Finally, the reports demonstrating the potential benefits of non-therapeutic approaches, such as calorie restriction, diet modification, and special nutritional interventions, including probiotics, prebiotics, synbiotics, and fecal microbial transplantation (FMT), for the treatment of GD-associated depression were collated.

## 2. Gut Microbiota

### 2.1. The Composition, Dynamics, and Functions of Gut Microbiota

The term GM refers to trillion of microorganisms (including bacteria, archaea, micro eukaryotes, fungi, yeast, and viruses) residing in the GI tract, exhibiting a commensal, symbiotic, or pathogenic relationship, which plays a crucial role in the host’s homeostasis [8,24,25]. The gut microbiome is composed of more than 10^14^ microorganisms, comprising a total mass of 0.3% of an individual’s body weight and coding for genomes of approximately 450 times higher when compared to the human genome [26]. The microbiota comprises approximately 1100 predominant and 160 other species types. The major phyla in human GM are Bacteroidetes and Firmicutes, while minor phyla include Proteobacteria, Actinobacteria, Verrucomicrobia, and Fusobacteria [14,27]. There are three primary enterotypes: (i) genus Prevotella (anti-inflammatory or protective), (ii) Bacteroides (pro-inflammatory and linked to the metabolic diseases and chronic pathological conditions), and (iii) Ruminococcus [28], whose relative abundance play a role in GD, including in depression. Collectively, both the diversity and composition of the intestinal microbiota vary dynamically from birth until the 5th year of age, and then it seems to acquire relative stability and becomes essentially comparable to that of an adult [29,30,31]. Furthermore, this dynamic system in the GI tract is altered by both endogenous (sex, age, race, ethnicity, and genetic makeup) and exogenous factors (physical exercise, diet, exposure to infections or diseases, use of antibiotics, and drug abuse). A microbiome metagenomic sequencing study conducted in infants revealed that fecal microbiota composition is individual-specific [10,32]. The reason for the microbiota diversity has not yet been fully understood; nevertheless, diet (including maternal milk), environment, comorbidity conditions, host genetics, and early life infections were speculated to be involved [33].

Maintaining a healthy GM improves the intestinal epithelial barrier integrity, stimulates the intestinal cell regeneration, produces mucin, and nourishes mucosa through elevated levels of short-chain fatty acids (SCFAs) [34]. The GM and enteroendocrine cells (EECs, a special type of intestinal epithelial cells that account for <1% of the epithelial population) synergistically regulate digestion, growth, and immune defense [35]. GM also shapes the innate immune system by regulating the maturation of intestinal-related lymphoid tissue in the early stages of life and the activation of the acquired immunity by stimulating local and systemic inflammatory responses [36]. The GM promotes the secretion of antimicrobial peptides by epithelial cells and reinforces tight junctions [37], regulates the metabolism of hormones, specific micronutrients, and vitamins, and also plays a crucial role in the absorption, metabolism, and elimination of xenobiotics.

### 2.2. Microbial-Derived Metabolites and Their Physiological Functions

The GM produces several functionally important metabolites, such as SCFAs (acetic acid, propionic acid, butyric acid, hexanoic acid, pentanoic acid, isovaleric acid, and isobutyric acid), branched-chain amino acids, LPS, bile acids, and catecholamines, and synthesizes or induces the secretion of neurotransmitters, including tryptophan precursors and metabolites, 5-HT, melatonin, glutamate, GABA, histamine, dopamine, and noradrenaline [8,24]. In addition, the GI tract is known to release various gut hormones, such as peptide YY (two members of the pancreatic polypeptide (PP-fold) family, which induces satiety and improves glucose homeostasis and behavioral functions), neuropeptide Y (which regulates the activity of the immune system, anxiety, and mood), leptin, ghrelin, cholecystokinin, and glucagon-like peptide-1 (GLP-1). These gut-derived metabolites are secreted either directly or indirectly by the GM, interacting with the CNS and enteric the nervous system (ENS). Dietary fibers such as complex carbohydrates are fermented by anaerobic microbes into SCFAs in the gut (Figure 1) [27,28,38]. Indeed, SCFAs modulate the levels of neurotransmitters such as glutamine, glutamate, and GABA in the hypothalamus and promote anorexigenic neuropeptide expression. SCFAs also upregulate the expression of tryptophan 5-hydroxylase 1, a key enzyme for 5-HT synthesis, and also tyrosine hydroxylase, a rate-limiting enzyme in the biosynthesis of dopamine, adrenaline, and noradrenaline [39,40].

The GM exhibits a symbiotic relationship with EECs and common subtypes, including A, K, and L cells. In response to chemical and mechanical stimuli, EECs release more than 20 different types of signaling molecules. These molecules reach the nucleus tractus solitarius and the hypothalamus in the CNS, and act locally or activate the afferent vagal terminals in the gut or liver [41,42]. In addition, SCFAs stimulate EECs to releases neuropeptide Y, peptide YY, and other gut hormones such as ghrelin, which attenuate anxiety- and depression-like behavior [41,43,44,45]. Bile acids are endogenous molecules synthesized in the liver, and their concentration and composition are significantly affected by food intake, particularly fat consumption, which in turn decreases downstream metabolism by the GM. Bile acids activate the nuclear receptor farnesoid X (FXR) in the ileum, resulting in the generation of fibroblast growth factor 19 or its ortholog fibroblast growth factor 15, which reach the systemic circulation and penetrate the blood–brain barrier (BBB) in mice, which suppresses the HPA axis and prevents depression [46,47,48]. Intestinal endocrine L cells express the G protein-coupled bile acid receptor (TGR5), and are influenced by microbial activity. TGR5 signaling regulates glucose homeostasis through increased GLP-1 production by L cells. The farnesoid X receptor, which controls GLP-1 production, is also expressed in the L cells [49,50,51], which regulate the glucose metabolism and prevent a depressive condition by balancing the HPA axis. Indole is another signaling molecule produced by gut bacteria from tryptophan via enzyme tryptophanase which modulates the secretion of GLP-1 by L cells, and induces the genes promoting tight junction resistance and anti-inflammatory cytokines (e.g., IL-10 and IL-4) in the human HCT-8 cell line derived from enterocytes. Indole is further metabolized into oxidized and conjugated derivatives such as oxindole and isatin, which are known to possess neuro-depressant properties [52]. Thus, gut microbial-derived metabolites also play a crucial role in regulating the hormonal, immune, and neural systems.

### 2.3. Gut–Brain Axis or Gut–Microbiota–Brain Axis

The brain and gut microbiome communicate bidirectionally via major pathways such as the vagus nerve, neuroendocrine, neuroimmune, autonomic nervous system (ANS), ENS, and the humoral links, otherwise termed as GBA [53,54,55]. Through neuroendocrine and enteroendocrine signaling pathways, microbes in the GI tract and their metabolites communicate with the CNS and ENS [41], as well as with biological barriers such as the intestinal mucosal barrier and the blood–brain barrier [56]. The next major communication route is directly mediated through parasympathetic and sympathetic components of the ANS and through ENS (mainly HPA) [57]. Vagal afferent pathways play an important role in HPA activation, which coordinates the adaptive responses to a variety of stressors like emotional and environmental stresses. Additionally, the GM can act on the vagus nerve [58] by producing bacterial metabolites via tryptophan metabolism [59,60]. HPA activation leads to the stimulation of the immune system with the increased production of pro-inflammatory cytokines and chemokines [61]. Together, these findings highlight the role of the GM and its metabolites in establishing GBA communication and maintaining the host’s homeostasis.

## 3. The Pathological Mechanisms Underlying GD-Associated Depression

### 3.1. Gut Dysbiosis in Depression

GD is defined as an abnormal alteration in the composition and diversity of the GM (particularly with an increase in the relative abundance of pathogenic microbes, i.e., the pathobionts, and a decrease in the relative abundance of beneficial microbes, i.e., symbionts), affecting the GI tract and neural and immune homeostasis [9,62]. GD directly affects the synthesis of neurotransmitters such as 5-HT, dopamine, glutamate, noradrenaline, and GABA in the gut lumen, while vice versa alterations in these neurotransmitters affect the microbial composition and abundance [14]. The pathological mechanisms underlying GD-induced depressive symptoms include: altered microbial composition, abundance, and metabolites, breakdown of the intestinal barrier integrity (reduced expression of tight junctions proteins such as claudin-5 and occludin in the GI tract), loss of goblet cells (resulting in reduced mucus secretion and thinning of the mucus layer), and translocation of pathobionts and toxic metabolites into the blood circulation, leading to chronic local and systemic inflammatory responses. GD can represent microbe-associated molecular patterns (MAMPs) that constitute bacterial products, including flagellin and LPS [63]. MAMPs in turn stimulate NLRP3 inflammasome and NF-kB, which are recognized by pattern recognition receptors of the innate immune system, leading to the increased production of pro-inflammatory cytokines (e.g., interleukin (IL)-18, IL-1, IL-6, and TNF-α) and peptidoglycan metabolites [64,65]. LPS activates the toll-like receptor (TLR)-4 and peptidoglycan stimulates nucleotide-binding oligomerization domain-containing protein-1 and/or nucleotide-binding oligomerization domain-containing protein-2 [66,67,68], which is linked to depressive behavior. Further, LPS translocates from the gut to the brain via the leaky mucosal barrier and negatively affects brain functions by disrupting the blood–brain barrier with decreased levels of tight junctions and anchoring junction proteins in the frontal cortex, hippocampus, and striatum [67,69]. These data indicate that GD affects neurochemical signaling and initiates the cascade of pro-inflammatory pathways, which are positively linked with depressive behavior [70].

### 3.2. GBA Dysregulation in Depression

Most importantly, the bidirectional dialogues between the gut and the brain via the HPA axis are affected significantly in depressive experimental models and patients, which are summarized below. GD-induced shifts in SCFA composition and amount reduce the 5-HT levels by modulating the tryptophan metabolism and induce depressive-like symptoms. The levels of acetic and propionic acids are decreased, while the levels of isocaproic acid level were significantly elevated in depressive women compared to healthy controls [71]. Moreover, the excessive production of indole by GD elevated the levels of oxindole and isatin, which induce hypotension, depressive-like helplessness behaviors, and cognitive impairment [57].

### 3.3. HPA Dysregulation in Depression

#### 3.3.1. Stress-Induced Activation of HPA

The HPA axis is one of the most essential components of GBA, providing a main biological response to stressful stimuli [72]. Environmental, emotional, and physiological stress is shown to increase systemic pro-inflammatory cytokine levels, which stimulate the HPA axis by triggering the paraventricular nucleus of the hypothalamus to secrete the corticotropin-releasing hormone (CRH) [53,73]. Increased CRH levels activate the secretion of the adrenocorticotrophic hormone (ACTH) from the anterior pituitary. In turn, ACTH leads to the release of glucocorticoids from the adrenal cortex. Excessive cortisol levels induce GD, which disrupts intestinal permeability, resulting in leaky gut and endotoxemia (an increase in immunogenic LPS released from Gram-negative bacterial cell wall) [58,74,75,76]. Increased levels of LPS stimulate gut inflammation and neurodegeneration via the hyperactivation of HPA, leading to the over-production of cortisol, which is linked to depression [55]. HPA hormones act in a negative feedback loop, in which cortisol lowers its own secretion by signaling the hypothalamus and pituitary gland to reduce the synthesis of CRH and ACTH [77,78]. Cortisol secretion is associated with circadian regulation (high levels in the morning and low levels in the middle of the night). Intriguingly, circadian rhythm and GM develop parallelly during the first 6 months of life [79]. The results from animal studies have supported that germ-free (GF) mice experience hyperactivity of the HPA axis with decreased *Stat5* gene expression in the hippocampus and increased corticotrophin-releasing factor (CRF) expression in the hypothalamus. Depressed mice were found to have an increased abundance of Actinobacteria and decreased Bacteroidetes in the gut, resulting in an aberrant immune tolerance and behavior, and higher levels of serum corticosterone [80,81]. Nobuyuki Sudo et al. (2004) reported that in GF mice, even moderate restraint stress leads to an increased corticosterone and ACTH release compared to unstressed GF mice [82]. Stress-induced changes in GM composition promoted the growth of Gram-negative bacteria, leading to increased intestinal permeability and the translocation of bacterial components across the intestinal lumen through inflammatory responses. These inflammatory mediators, such as pro-inflammatory cytokines (IL-1, IL-6, and TNF-α) and prostaglandins, hyperactivated HPA, which further stimulated the microbiota–immune–neuroendocrine interactions, resulting in the precipitation of depressive behavior [82].

#### 3.3.2. LPS- and Peptidoglycan (Derived from Pathogens)-Induced Activation of HPA

The HPA axis is activated by LPS and peptidoglycan (which act as MAMPs) derived from pathogens and the related production of pro-inflammatory mediators. Peptidoglycan, a cell wall component of most bacteria, crosses the blood–brain barrier and activates particular pattern recognition receptors of the innate immune system, negatively affecting brain development and behavior. Further, gut bacteria-derived LPS and peptidoglycan stimulate innate immunity by activating nucleotide-binding oligomerization domain-containing protein-1 (NOD1) [66]. LPS and TLR-4 activators cross the intestinal epithelial barrier in response to physical immobilization stress or a Western-style high fat diet exposure, resulting in the hyperactivation of inflammation and the HPA axis. A TLR-2 agonist by lipoteichoic acid from *Bacillus subtilis* leads to an activated immune system in the periphery and brain, contributing to the upregulation of the cytokines expression in the amygdala and prefrontal cortex, with decreased expression of tight junctions-associated proteins in the brain. These changes, in turn, increase the plasma concentration of corticosterone, resulting in the hyperactivation of the HPA axis, which has a negative impact on emotional-affective behavior, including anxiety and depressive conditions in mice [83].

### 3.4. Chronic Oxidative and Nitrosative Stress in Depression

Chronic HPA activation stimulates the inflammatory pathways and increases the formation of reactive oxygen species (ROS) and reactive nitrogen species (RNS), which can damage DNA, lipids, and proteins in mitochondria and cell membranes [84]. The prevalence of oxidative and nitrosative/nitrative stress is reflected by the elevated amounts of lipid peroxidation by-products, such as malondialdehyde and 4-hydroxynonenal, in depressed patients [85]. Additional reports show significant decreases in endogenous antioxidants, such as melatonin, glutathione, glutathione peroxidase, etc. (Figure 2), which are protective in the mitochondrial function and are involved in the regulation of cAMP/circadian genes, whose dysregulation may lead to behavioral disorders [78,86,87]. GF mice have lower antioxidant enzymes (e.g., catalase, glutathione peroxidase, and superoxide dismutase) with increased NADPH oxidase and nitric oxide production, resulting in oxidative damage [88]. Pathobionts and/or pathogenic taxa lead to depression either directly (by generating valeric acid, an inverse agonist at the adenosine A1 receptor) or indirectly (by increasing the synthesis of kynurenine from tryptophan) [89]. Further, the translocation of pathobionts from the gut to mesenteric lymph nodes activates monocytes and macrophages, contributing to the elevated production of inflammatory mediators and oxidative stress [90]. Intriguingly, the increased translocation of Gram-negative bacteria or bacterial components (e.g., LPS or peptidoglycan) from the gut into the blood stream elevates IgM and IgA levels, leading to hyperimmune responses by binding to TLR-2/4 complexes. GD-induced immune activation increases pro-inflammatory cytokines and ROS/RNS, thereby resulting in oxidative stress [21]. The latter, in turn, generates redox-derived DAMPs, which further activate TLR-2/4 complexes, resulting in a vicious inflammatory loop known as the TLR-2/4 radical cycle, which is suggested to be a significant causative factor for chronic immunological activation and oxidative/nitrosative stress in various neuro-immune diseases [78,91]. In experimental animal studies, LPS injections resulted in higher amounts of malondialdehyde, nitrite, and nitrate in the brain, as well as lower levels of glutathione, a finding observed in many psychiatric illnesses [92,93]. Furthermore, the translocation of gut bacteria may cause the formation of neo-antigenic determinants in MDD patients, leading to the disease’s proclivity for autoimmunity [85]. LanCL1, a glutathione-binding protein, participates in antioxidant activities, protecting neurons, and thus may have a defensive role in neurodegenerative diseases. At 9 weeks of age, the expression of the tight junction proteins (ZO-1 and occludin) in LanCL1-knockout mice were significantly decreased, with increased circulatory pro-inflammatory cytokines (IFN-γ, TNF-α, IL-1β, and IL-6), which might be linked to gut leakiness with increased systemic oxidative stress. Further, propionic acid was found to be significantly elevated in the knockout mouse feces, where propionic acid was known to induce neuroinflammation, triggering cytokines, such as IL-8, IL-1, IL-6, and TNF-α. These changes, in turn, stimulated the production of acute phase proteins, such as haptoglobin and the C-reactive protein (a well-known inflammatory marker), and oxidative stress, which activated a transcription factor NF-kB, leading to neuronal damage associated with depressive- and anxiety-like behaviors [14,94].

### 3.5. Altered Metabolism of Serotonin and Tryptophan in Depression

The production of several neurotransmitters such as 5-HT, norepinephrine, GABA, and dopamine are directly regulated by GM [65,95]. The GI tract contains high concentrations of 5-HT (and melatonin). Kim and Camilleri found that 90% of 5-HT is secreted by epithelial ECCs, with the remaining 10% from the ENS [96]. Several microorganisms produce neurotransmitters, such as acetylcholine (e.g., *Lactobacillus plantarum*), dopamine (e.g., *Proteus vulgaris*, Bacillus, and *Serratia marcescens*), GABA (e.g., Lactobacillus and Bifidobacterium), histamine (e.g., Citrobacter and Enterobacter), norepinephrine (e.g., Saccharomyces, Bacillus, and *E. coli*), and 5-HT (e.g., *Escherichia coli*, Enterococcus, Candida, and Streptococcus) [97]. GD depletes tryptophan concentration in plasma and excessively activates the immune system. IFN-γ enhances the activity of indoleamine 2,3-dioxygenase enzyme, converting tryptophan to formyl-kynurenine, which is strongly correlated with MDD [98,99]. The 5,6,7,8-Tetrahydrobiopterin is a co-factor of various amino acid converting enzymes responsible for the biosynthesis of neurotransmitters such as tryptophan, 5-HT, dopamine, and noradrenaline. GD causes chronic low-grade inflammation, which decreases 5,6,7,8-tetrahydrobiopterin synthesis and thus affects the production of neurotransmitters [100,101]. Within the GBA pathway, GM either directly or indirectly recruit tryptophan and facilitate serotonergic signals to control the tryptophan metabolism [95]. Tryptophan hydroxylase-1 is the rate-limiting enzyme for 5-HT synthesis in ECCs, and tryptophan hydroxylase genes 1 and 2 (i.e., *Tph1* and *Tph2*) are associated with the serotonergic system. Tph1 is mainly expressed in the periphery and Tph2 in the brain. Preclinical findings showed that a loss of functional single-nucleotide polymorphism in the human *TPH2* gene was observed in some patients with unipolar major depression and that over-expression of this mutant gene in PC12 mouse cells resulted in markedly decreased 5-HT levels compared to those with WT. However, there were no differences in dopamine levels, supporting a selective decrement of 5-HT levels by a mutant *TPH2* gene [102]. Khan and his coworkers (2019) recently reported that *Trp1*-knockout mice had altered GM profiles compared to those of the WT mice, while 5-HT can directly modulate the growth of commensal bacteria in vitro in a concentration and species-specific manner [103]. In addition, GF mice colonized with *Tph1*-knockout mice were resistant to dextran sulfate sodium-induced colitis, with a reduced 5-HT level in the gut, supporting the connection between GBA and 5-HT levels [103]. Furthermore, 5-HT increased the production of cytokines in the myeloid dendritic cells in vitro and *Tph1*-knockout mice showed a reduced concentration of pro-inflammatory cytokines (IL-1β, IL-6, and TNF-α) [104]. A clinical study supports that single-nucleotide polymorphism in the human *TPH2* gene with a constitutively suppressed function resulted in an 80% decreased production of 5-HT in the brain, which is connected to mood disorders and psychiatric symptoms such as suicidal behavior [105]. Clarke et al. (2013) reported that GF mice have elevated tryptophan levels, which are associated with increased 5-HT levels in the hippocampus. However, a lower *5-HT1A* receptor gene expression in the dentate gyrus was observed in female mice, but not in males. This is likely to explain sex-dependent CNS modulatory effects such as anxiety and depression, possibly through modulating the composition and abundance of gut microbes [16]. The amygdala and hippocampus of GF mice showed lower concentrations of 5-HT, BDNF, and the specific 5-HT1A receptor, all of which are associated with anxiety- and depressive-like behavior [16,106,107,108]. The administration of *Lactobacillus johnsonii* suppressed the activity of indoleamine 2,3-dioxygenase and reduced serum kynurenine concentrations by 17%, leading to increased 5-HT levels by 1.4-fold and the improvement of depressive behavior in rats [109,110]. These results indicate a complex interaction between GD and 5-HT-related neurotransmission in depression [111].

### 3.6. Altered Metabolism of Homocysteine in Depression

Homocysteine, an amino acid, is involved in one-carbon metabolism and is produced from the breakdown of folate, vitamin B6, and vitamin B12 [112]. Increased levels of homocysteine are found to be associated with various neuropsychiatric disorders, including depression [113,114]. The causative factors for hyper-homocysteinemia include folate deficiency and mutations in genes regulating homocysteine metabolism [115]. Altered homocysteine metabolism is found to be associated with mental retardation and psychiatric symptoms [115]. Mounting evidence has shown that folate deficiency and the impaired metabolism of 5-HT, dopamine, and noradrenaline lead to depression [116,117]. Clinical reports also indicate the presence of folate deficiency in depressed patients [112,118,119].

A genetic variant of the human methylenetetrahydrofolate reductase gene is associated with impairment of its ability to process folate, and this defective gene leads to increased levels of homocysteine [116]. Hyper-homocysteinemia can also result from changes in the vascular endothelium and is considered a risk factor for several disorders associated with chronic inflammation and depression [113,114]. Elevated levels of homocysteine can increase intestinal permeability and disrupt the epithelial barrier in rats by stimulating inflammatory and oxidative damage, which was ameliorated by probiotic administration [120]. Clinical studies report a high prevalence of homocystinuria in patients with psychiatric disorders. Treating homocystinuria patients with vitamin B6 has been shown to improve psychiatric conditions, including episodic depression, chronic disorders of behavior, and personality disorders [120,121]. In a recent report, pathogenic bacteria such as *Subdoligranulum sp., Eubacterium sp*., and *Clostridiales family XIII* were identified as the main producers of homocysteine [122], supporting the pathogenic role of homocysteine in GD and depression. As a consequence, vitamins and essential amino acids are a critical part of the healthy diet, while the GM plays an important role in ensuring that the host receives a steady supply of proper, well-balanced nutrients [121].

### 3.7. Neuroinflammation in Depression

GD-mediated gut leakiness with elevated LPS, immune dysregulation, and chronic inflammation play a significant role in the onset, severity, and resistance to treatment in depressed patients [123]. Elevated levels of proinflammatory cytokines, such as IL-6, IL-8, TNF-α, INF-γ, C-reactive protein, and IL-1β [15], due to peripheral chronic inflammation and central microglial dysfunction, are associated with depressive symptoms at both the onset and during increased severity. Using GF mice, Erny et al. (2015) demonstrated that GM has a role in the onset of depression, particularly during early age, through the modulation of the microglial-associated immune network in the brain [124]. Supportively, gut microbial-derived metabolites, such as SCFAs, are involved in the maturation and immunological functions of microglial cells in the CNS [124]. Higher cytokine levels activate the HPA and disrupt the serotonergic and noradrenergic circuitries to deplete 5-HT levels and increase the indoleamine 2,3-dioxygenase enzyme [125]. MDD is associated with elevated inflammasome, which activates the microbial endogenous signal via pattern recognition receptors, which, in turn, stimulates caspase-1, promoting the release of pro-inflammatory cytokines such as IL-1β, IL-6, and IL-18. Elevated cytokines can increase glucocorticoid resistance, and negatively affect the synthesis and metabolism of 5-HT, norepinephrine, and dopamine [126]. Abnormal microglial dysfunction was observed in the brain samples at the onset of depression [127]. Microglia in depressed patients or patients under chronic stress become increasingly dysfunctional and lose neuroprotective properties. Hyperactivated microglia negatively affect neuroimmune homeostasis by overexpressing proinflammatory cytokines (e.g., TNF-α and IL-1β), class I and II major histocompatibility complex antigens, and neurotoxic molecules (e.g., RNS/ROS, superoxide anions, large amounts of nitric oxide, peroxynitrite, etc.), all of which lead to altered neuroinflammatory pathways, resulting in depression [19]. Anti-caspase-1-treated mice were shown to have increased Akkermansia and Blautia species with the induction of Foxp3 T-regulatory cells as well as the suppression of IL-1β- and IL-6-mediated pathways [24,128]. GD is reported to promote the ingress of various peripheral immune cells, including CD4+ and CD8+ T cells, B cells, natural killer cells, neutrophils, and monocytes, into the brain, contributing to elevated neuroinflammation [129]. Bacterial endotoxins, including LPS, also stimulate intestinal and blood–brain barrier permeability, change neurotransmitter levels, increase IgA and IgM, and enhance both systemic and CNS inflammation [107,130]. Recent evidence has shown that neuroinflammation induced or reinforced by GD plays a crucial role in depression pathology, mainly by increasing cytokine release (peripherally by immune cells and centrally by microglia) and activating the HPA. These changes lead to elevated levels of glucocorticoids, systemic inflammation, cell-mediated immune activation, and neuronal apoptosis, and decreased 5-HT levels, neurogenesis, and neuroplasticity, all of which contribute to the development of depression [131,132].

## 4. Alterations in the Gut Microbial Abundance in Depression

### 4.1. Preclinical Evidence Using Germ-Free and Specific-Pathogen-Free Animal Models

Studies using GF animals (animal models devoid of complex GM) have provided better insights on the interaction between GM and neurobehavior of the host [16]. GI inflammation induces anxiety-like behavior and also affects CNS functions [133]. Proinflammatory markers, such as CCR2, lipoteichoic acid (which induces the secretion of cytokines including IL-1β and TNF-α that can contribute to the blood–brain barrier disruption), IL-22ra2, and nuclear factor-kB (NF-kB), are associated with the pathophysiology of inflammatory bowel syndrome and play a key role in anxiety-like behavior in a murine model of chronic social stress [68]. Similarly, in a mouse prenatal stress model, increased Rikenellaceae abundance was shown to correlate with decreased BDNF levels in the amygdala of female offspring. On the other hand, a decrease in Bifidobacteria was correlated with elevated proinflammatory cytokines and the upregulation of immune regulating genes. These changes trigger systemic inflammatory responses, which lead to the development of anxiety [134]. Additionally, from the fetal developmental until the adulthood stage, GF mice showed a lower expression of tight junction proteins, claudin-5, and occludin, which is linked to the decreased blood–brain barrier integrity. However, recolonization using healthy microbiota, especially with SCFAs-producing bacterial strains, are found to maintain the blood–brain barrier integrity [135]. Early life stress is reported to increase the abundance of Oscillibacter, Parasutterella, Treponema, Ruminiclostridium, and Helicobacter, and decrease the amount of Bacteroides, Rikenellaceae, *E. ruminantium*, Lactobacillus, and Parabacteroides. These changes are correlated to elevated levels of corticosterone, ACTH, and glucocorticoids receptors in the hippocampus. The altered gut microbial composition and abundance suppress miR-124a and increase miR-132 expression. Additionally, the increased expression of α-amino-3-hydroxy-5-methyl-4-isoxazolepropionic acid receptor (GluR1 and GluR2) and N-methyl-D-aspartate receptor (NR2A and NR2B) leads to elevated levels of glucocorticoids receptors in rats [11]. These changes in the gut microbiome and glutamate receptors are found to be positively correlated with the pathogenesis of depression. In GF mice, stress is induced via elevated ACTH, and corticosterone in plasma is partially reversed by colonization with the fecal microbiota obtained from specific pathogen-free animals, while it is entirely reversed by introducing a single organism, *Bifidobacterium infantis*, a common probiotic organism and a major bacterium in the infant gut [82]. Interestingly, GF mice have shown reduced anxiety-like behavior [82], while GF rats show an opposite phenotype and increased anxiety-like behavior [16]. Balb/C mice are highly vulnerable to stress, which is further confirmed by the FMT of Balb/C into GF Swiss (Webster SW) mice, resulting in increased anxiety behavior when compared to conventional SW mice [136]. These data strongly indicate the impact of colonization with specific strains of bacteria on the neurobehavioral changes, including anxiety [16,80,137]. Bruce-Keller and his team [136] reported that mice fed with a Western-style high-fat diet showed selective disruption in GM with increased Gram-negative Proteobacteria, activated TLR-2 and TLR-4, increased lymphocyte expression of Iba-l and matrix metalloproteinase 9, and decreased expression of endothelial tight junction proteins ZO-1 and Claudin-5. A Western-style high-fat diet also suppressed the abundance of a beneficial strain *Akkermansia muciniphila* (about 5.4-fold), contributing to elevated endotoxin levels in the serum and prefrontal cortex, resulting in neuroinflammation, while an increased corticosterone level leads to anxiety- and depression-like behavior in mice fed with a Western-style high-fat diet [138]. Preclinical studies using GF mice clearly indicated the positive association between GD and the development of depressive-like symptoms, and FMT techniques further bolster the causal relationship between microbes and improvement in depressive symptoms.

### 4.2. Clinical Evidence

Several clinical studies link the pathogenic influence of GD to depression and anxiety behaviors [60,62,139,140]. Disturbed GI symptoms, such as bloating, nausea, vomiting, abdominal pain, and constipation, are frequently reported in depressive patients [69]. Patients with anxiety or psychological distress are more often reported with the symptoms of irritable bowel syndrome (IBS) [133,141,142]. Disproportionate abundance of Faecalibacterium species (reduced), and Enterobacteriaceae and Alistipes species (increased), are correlated with depression pathology [143]. The presence of pathogenic microbiota families, such as *Ruminococcaceae, Shewanellaceae, Halomonadaceae,* and *Verrucomicrobiae,* were positively correlated with the anxiety-like behavior in patients, whereas *Lachnospiraceae* and *Bacteroidaceae* families were associated with both anxiety and IBS patients [53]. Individuals with GD were reported to have decreased 5-HT levels, slow bowel movements, and constipation. Jiang et al. (2015) reported that Bacteroidetes and Proteobacteria were more abundant in MDD patients, whereas the decreased abundance of *Lachnospiraceae* and *Ruminococcaceae* families, within the phylum *Firmicutes*, were observed in MDD patients when compared to a healthy group [10]. The microbial phylum (*Firmicutes*) is responsible for the production of SCFAs, which primarily prevent gut leakage and the secretion of proinflammatory cytokines [10] (Figure 3). Another study in young Americans with MDD reported lower levels of anti-inflammatory cytokines (such as IL-10) and butyrate-producing bacteria phylum *Firmicutes* and *Ruminococcaceae*. These alterations, in turn, caused intestinal barrier dysfunction and chronic low grade inflammatory responses. In this case, increased serum levels of IL-1β, IFN-α2, and IFN-γ are associated with MDD and this fact further adds a clinical link between GD and depression-like behavior [55]. More clearly, the clinical evidence also showed that fecal SCFAs concentrations are lower in depressed patients when compared to those of healthy controls [71]. These clinical findings strongly suggest the potential interlink between GD and depression-like behavior.

## 5. The Alternative Strategies Targeting Gut Dysbiosis-Associated Depression

There are many synthetic pharmaceutical agents in treating depression in humans, although their efficacies vary due to various factors [63]. In addition, restoring the healthy gut microbial composition and diversity (termed as eubiosis) via non-pharmacological approaches such as calorie restriction, intermittent fasting, ketogenic diets, and special nutritional interventions (e.g., probiotics, prebiotics, postbiotics, synbiotics, and FMT) are the recent alternative potential strategies popularly considered. Additionally, there are several instances of preclinical and clinical data that support the notion that the alternative approaches of taking prebiotics, probiotics, and synbiotics are likely to cause eubiosis, and that these approaches can also be considered as a potential strategy for improving depression and other stress-related disorders [144,145,146].

### 5.1. Probiotics

Probiotics consist of live microorganisms, which stimulate the growth of beneficial microbes, when given in appropriate amounts [147]. The term “psychobiotics” is also used to denote the probiotics that are used widely in the treatment of neuropsychiatric disorders, because of their neuroprotective properties [148]. In mice subjected to stress, feeding with probiotic *Lactobacillus rhamnosus* (JB-1) downregulated the stressed-induced GABA Aα2 mRNA expression in the prefrontal cortex and amygdala, and reduced the corticosterone levels. However, it did not influence GABA Aα2 expression in the hippocampus [23,111]. Probiotic *Lactobacillus farciminis* treatment prevented the gut barrier leakiness and reversed psychological stress-induced HPA axis activation in rats [149]. Administration of a probiotic such as *Bifidobacterium longum* normalized the hippocampal BDNF levels and reduced the inflammation-induced anxiety-like behavior in animal models [111,150]. However, in IBS patients, probiotics are found to reduce only depressive-like behavior, but not anxiety [146]. Early-life stress in rats downregulated miR-124a and up-regulated miR-132 expression, and interfered with glucocorticoids receptors expression, which led to anxiety and depressive behavior. The supplementation with Lactobacillus reversed the expression of miR-124a/132 [11]. The Oscillibacter strain produced a metabolic end-product, i.e., valeric acid, which binds to the GABA–A receptor and releases GABA and helps to treat anxiety and insomnia [151,152,153]. Supplementation with *Bifidobacterium longum* 1714, a probiotic strain, was found to be effective in decreasing stress-induced cortisol release and lowering the daily self-reported stress levels by the improvement of visuospatial memory in healthy male volunteers [154]. In another study, a cocktail of probiotics, including *Bifidobacterium lactis* W52, *Lactobacillus acidophilus* W37, *Bifidobacterium bifidum* W23, *Lactobacillus brevis* W63, *Lactobacillus salivarius* W24, *Lactobacillus casei* W56, and *Lactococcus lactis* (W19 and W58), has shown promising results in reducing negative thoughts and behavior [12]. These findings from experimental and clinical trials reveal the beneficial effects of live probiotics in the treatment of GD-associated depression.

### 5.2. Prebiotics

Prebiotics refer to specific substrates (non-digestible fibers such as oligosaccharides) that promote growth and functions of specific beneficial gut microbes [155]. Dietary prebiotics such as fructo-oligosaccharides and B-immuno galacto-oligosaccharide stimulated the growth of beneficial bacteria such as *B. longum* and reduced the stress-induced activation of the HPA axis in healthy young volunteers [22,156,157]. B-immuno galacto-oligosaccharide administration has been shown to reduce LPS-induced anxiety- and depressive-like behavior in rats [158]. Crocin-I improved the GM composition and SCFAs levels, and reduced the leakage of LPS from the intestine into the systemic circulation, while it also increased BDNF levels in depressed mice [26]. Table 1 explains the beneficial effects of various probiotics and prebiotics in recent studies.

### 5.3. Postbiotics

Postbiotics are the inactivated microbial cells mixed with or without metabolites (e.g., lactic acid, proteins, vitamins, and SCFAs) or cell components (including pili, cell wall components), which confer health benefits to the host [168]. Similar to probiotics, the beneficial mechanisms mediated by postbiotics also include promoting the integrity of the epithelial barrier function, restoring the harmonical balance of the microbiota composition and diversity, regulating the immune responses, and regulating the gut–brain axis signaling [168,169]. In adolescent male C57BL/6J (B6) mice exposed to sub-chronic and mild social defeat stress, the administration of the heat-killed *Lactobacillus helveticus* strain MCC1848 reduced depression- and anxiety-like behaviors, and was shown to improve the genes involved in neuron differentiation and development and signal transduction in the nucleus accumbens [170]. Heat-killed *Enterococcus fecalis* (EC-12) fed along with diet reduced the depression- and anxiety-like behaviors, upregulated the neurotransmitter receptor *Adrb3* and *Avpr1a* genes expression in the prefrontal cortex, and also increased the relative abundance of Butyricicoccus and Enterococcus composition in the intestine of adult male C57BL/6 J mice [171]. In 241 healthy adult volunteers, two doses of heat-killed *Lactobacillus paracasei* MCC1849 before breakfast for 12 weeks increased resistance to the common cold and maintained a desirable mood state [172]. The intake of two tablets per day for 24 weeks of heat-inactivated *Lactobacillus gasseri* CP2305 (CP2305) reduced the anxiety, increased the sleep quality and n-valeric acid, and restored the microbial composition in young adults exposed to chronic stress [173].

### 5.4. Synbiotics

Synbiotics are the synergistic mixture of prebiotics and probiotics aimed to benefit the host by promoting beneficial microbial activity. Markowiak and Slizewska (2017) reported that the administration of synbiotics caused a significant decrease in the malondialdehyde and hydrogen-peroxide concentration in the human plasma [174]. Additionally, the administration of synbiotics resulted in significantly higher plasma levels of glutathione and free sulfhydryl groups in women compared with men. This gender variation could be explained by different concentrations of hormones, such as estradiol and testosterone, which are associated with psychological stress [175,176]. A recent randomized study by Haghighat et al. (2021) reported that synbiotics (*Lactobacillus acidophilus* T16, *Bifidobacterium bifidum* BIA-6, *Bifidobacterium lactis* BIA-7, and *Bifidobacterium longum* BIA-8) increased the serum levels of BDNF and improved the depression symptoms in depressed patients compared to controls [145].

### 5.5. Polyphenols

Polyphenols also improve the immune-modulating bacteria, such as Bifidobacteria and Lactobacilli (Table 2), as well as preventing the colonization of pathobionts [177], despite the very low bioavailability of these phytochemicals [178,179]. Table 2 summarizes the potentials of various polyphenols used in the management and/or treatment of depression.

### 5.6. Diet Modifications

Even a high fiber diet lowers colonic pH, which prevents the over-growth of pathogenic bacteria [191]. The administration of a modified amino acid diet (rich in lysine, histidine, and threonine, and low in valine, leucine, and isoleucine) was demonstrated to enhance repetitive self-grooming behavior in BTBR T+tf/J mice, while it simultaneously inhibited the mammalian target of rapamycin signaling in the prefrontal cortex, thereby facilitating autophagy [192]. A supplementary diet, containing omega-3 and omega-6 polyunsaturated fatty acids, regulates the microbial metabolism by reportedly protecting the microbial composition and abundance, especially during early life stress, and it increases both Bifidobacterium and Lactobacillus [193,194]. Furthermore, supplementation with omega-3 fatty acids, such as docosahexaenoic acid and eicosapentaenoic acid, has been shown to reduce the stress-induced activation of the HPA axis in adolescent female rats, and also improved the cognition in adulthood, indicating the stress- and depressive behavior-reducing benefits of a dietary supplement [194,195]. Numerous studies, both in humans (Table 3) and animal models, showed the potential benefits of prebiotics, probiotics, and special nutritional interventions in the treatment of depression-like behavior by targeting GD and depression via GBA [22,158,196]. Interestingly, many plants-based supplements exert beneficial effects on the gut microbe abundance and alleviate mental disorders, including depression (Table 2).

### 5.7. Fecal Microbiota Transplantation (FMT)

FMT is used to restore eubiosis by transferring the healthy microbiota (either complete or specific microbiome of the donor) to the gut of the recipients via endoscopy, enema, and oral feeding of freeze-dried materials. Similar to probiotics, FMT is established in order to achieve healthy intestinal microbial composition and function. FMT treatment showed beneficial effects as the transfer of fecal microbiota from healthy donors alleviated depression- and anxiety-like symptoms in mice subjected to stressful conditions. FMT has become one the most commonly applied methods in various GI and neuropsychiatric disorders [197]. Clinical studies have reported that IBS and GI tract-related problems are linked to depression. FMT from a healthy donor to the diseased patients alleviated depression- and anxiety-like behavior in IBS patients as well as *Clostridium difficile* infection in elderly people [70,198].

## 6. Conclusions

Depression is a debilitating disorder affecting millions of people every year, with high morbidity and fatality rates. MDD is characterized by either depressed mood and/or anhedonia, which are frequently associated with anorexia, psychomotor retardation, poor productivity, insomnia or hypersomnia, agitation, negative thinking, and suicidal thoughts. Recent studies have clearly shown a positive correlation between GD and the pathogenesis of depression. GD acts a risk factor for the development of MDD and/or worsens the severity of existing mental illness and GI disorders. Recent attempts to investigate the role of GD in the depression pathophysiology have highlighted that GD causes the dysregulation of the GBA, leading to the increased production of toxic microbial metabolites, intestinal leakage, endotoxemia, and the release of immune mediators. All these changes result in a chronic pro-inflammatory status, which in turn triggers changes in the blood–brain barrier, neurotransmission, neuroinflammation, and behavior.

Experimental models have also revealed that GD alters the intestinal immune homeostasis via HPA axis activation with the disruption of the intestinal barrier and bacterial translocation, which manifest as a leaky gut with elevated levels of LPS and the release of proinflammatory cytokines. GD also negatively affects the synthesis and/or metabolism of neurotransmitters such as norepinephrine, 5-HT, and dopamine, ultimately resulting in depression. In line with experimental animal models, recent clinical studies have shown that increased levels of proinflammatory biomarkers of GD negatively impact GBA signaling in psychiatric disorders such as depression and anxiety. Further, the levels of these pro-inflammatory markers were found to be positively associated with the severity of depression. Although the classical therapeutics such as antidepressants are used widely to treat depression, their efficacies vary, and adverse effects are unavoidable. Therefore, there is an urgent need for new therapeutic targets as well as the development of alternative approaches to alleviate or improve the severity of GD-associated depression and anxiety for many patients suffering from neuropsychiatric disorders.

When considering the findings discussed in this review, potential alternative strategies, such as specific nutritional interventions using prebiotics, probiotics, synbiotics, and diet modifications, FMT, and naturally derived compounds, such as omega-3 fatty acids, *S*-adenosylmethionine, and polyphenols, appear to be promising candidates. These agents or approaches are found to elevate 5-HT, dopamine, and BDNF levels with improved intestinal barrier function by regulating the GBA in depressive experimental models and in patients. Specifically, probiotics have produced promising results in depression treatment, and synbiotics are investigated widely for the beneficial outcomes on depressive symptoms. Furthermore, FMT seems to be significantly effective in treatment-resistant depression, which does not respond to the classical psychiatric drugs, mainly by reversing the GD, and establishing eubiosis. Indeed, most of the studies examining GBA are conducted in experimental animal models, and this warrants more pilot followed by large-scale randomized cohort studies in depressive patients.

## Figures and Tables

**Figure 1 cells-11-01362-f001:**
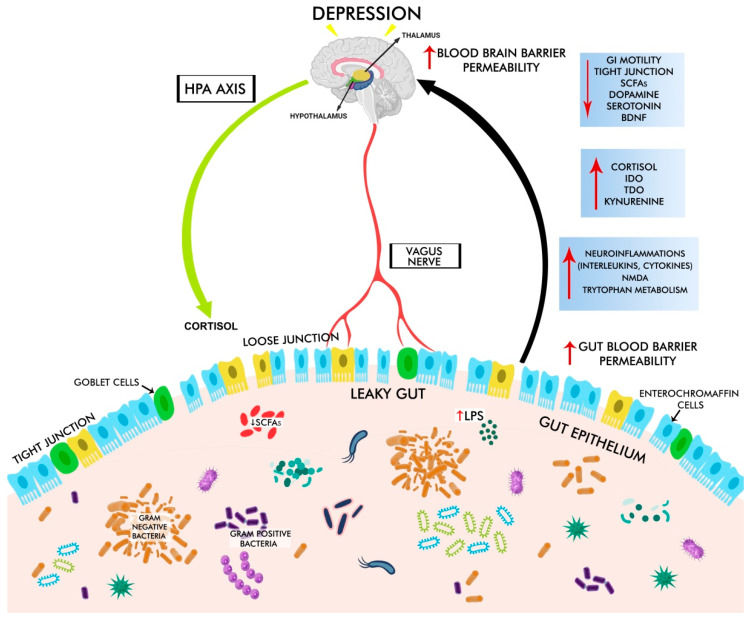
The gut–brain axis and depressive disorder. Gut dysbiosis increases intestinal barrier permeability, causing leaky gut with decreased levels of tight and adherent junction proteins, loss of goblet cells, increased lipopolysaccharide (LPS, an endotoxin derived from the Gram-negative bacterial cell wall), and increased blood–brain barrier permeability. These changes trigger the production of pro-inflammatory cytokines via HPA disruption and increased cortisol levels, which in turn leads to decreased levels of 5-HT and dopamine, BDNF, and IDO, TDO, kynurenine, resulting in depression.

**Figure 2 cells-11-01362-f002:**
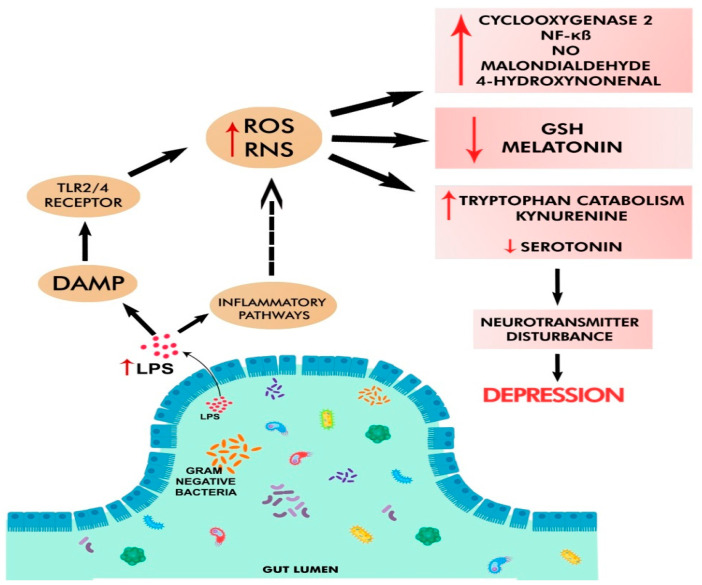
Oxidative and nitrosative stress in depression. Pathobionts in GD elevate gut leakiness and circulating LPS levels, which trigger DAMP, TLR2/4 receptor, and inflammatory pathways. These lead to increased levels of reactive oxygen/nitrogen species (ROS/RNS), cyclooxygenase-2 (COX-2), NO, malondialdehyde, 4-hydroxynonenal, tryptophan catabolism, kynurenine, and decreased amounts of antioxidants (e.g., glutathione, melatonin, and serotonin). All these changes negatively affect neurotransmission and cause depression.

**Figure 3 cells-11-01362-f003:**
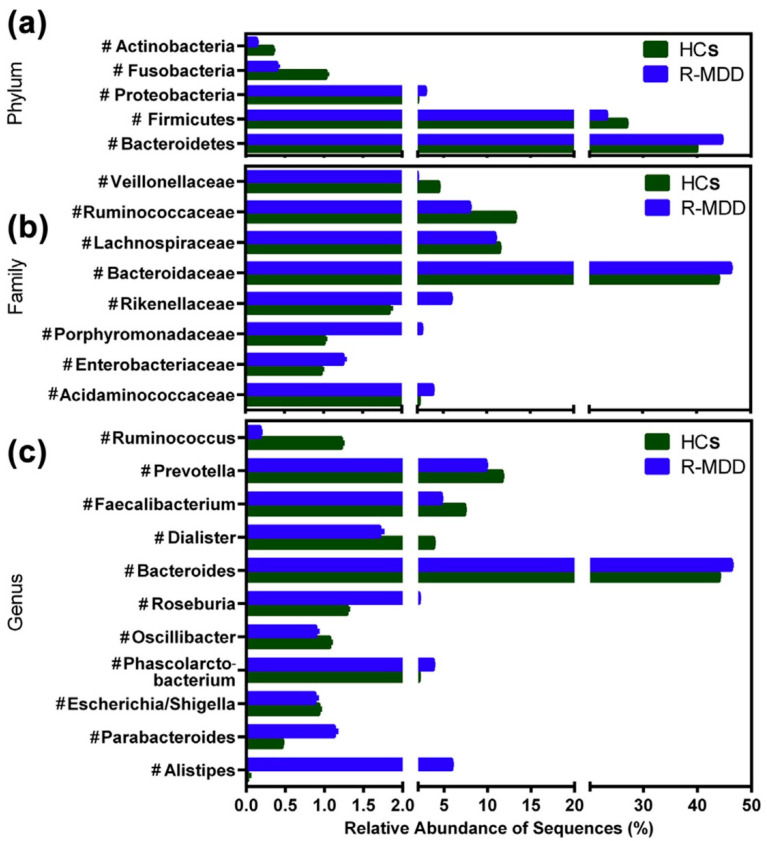
Taxonomic changes in fecal microbiota and relative abundance between healthy control (HC) and MDD groups at the levels of bacterial phylum (**a**), family (**b**), and genus (**c**). *Enterobacteriaceae, Fusobacteriaceae,* and *Rikenellaceae* were considerably higher, but *Prevotellaceae, Bacteroidaceae*, and *Ruminococcaceae* were significantly lower in MDD when compared to HC. *Alistipes, Blautia, Lachnospiracea, Parabacteroides, Clostridium XIX,* and *Oscillibacter* are more prevalent in MDD than in HC. *Prevotella, Bacteroides, Faecalibacterium,* and *Ruminococcus* are less abundant in MDD than in HC. The figure is reused as per journal copyright permission [10].

**Table 1 cells-11-01362-t001:** Examples of probiotics, prebiotics, and postbiotics used to treat anxiety- and depressive-like behavior.

Strains (Probiotics/Prebiotics/Postbiotics)	Outcome	References
*Lactobacillus helveticus* and *Bifidobacterium longum*	Normalizes hippocampal BDNF levels and inflammation	[111,150,159]
Fructo-oligosaccharides and B-immuno galacto-oligosaccharide	Stimulate the growth of beneficial bacteria, i.e., *B. longum,* which leads to reduction of stress-induced activation of the HPA axis, corticosterone levels, and pro-inflammatory cytokines, and increased BDNF	[22,146,157,160,161]
*Bifidobacterium infantis 35624*	Reduces depressive-like behavior via alleviating 5-HT	[82,109,162]
*Bifidobacterium breve*	Stimulates 5-HT receptors in intestinal cells of rats	[163]
*Bifidobacterium longum PS128*	Produces beneficial metabolites (SCFAs) and improves the locomotor activity in depression	[164]
*Lactobacillus plantarum*	Decreases stress-induced anxiety-like behavior	[164]
*Lactobacillus rhamnosus (JB-1)*	Reduces GABA Aα2 mRNA and corticosterone	[23]
*Lactobacillus farciminis*	Prevents gut barrier leakiness and reverses psychological stress-induced HPA axis activation	[149,165].
*Lactobacillus casei strain Shirota*	Reduces anxiety scores in patients with chronic fatigue syndrome and increases abundance of Lactobacillus and Bifidobacterium in fecal samples	[166,167]

**Table 2 cells-11-01362-t002:** Summary of the studies that indicate the potential anti-depressant effects of various polyphenols.

Source	Main Compound	Outcome	References
Cocoa	Catechins Anthocyanins Proanthocyanins Flavanols Epicatechin	Prevents neuroinflammation in the dorsal vagal complex	[180]
Blueberry	Anthocyanins	Significantly increases brain activity with improved working memory and depression-like behavior	[181]
Coffee	Flavanols Caffeic Acid, Chlorogenic Acid	Markedly increases cognitive performance, psychomotor control, and working memory	[182]
Strawberry	Fisetin (polyphenol)	Suppresses proinflammatory markers such as TNF-α	[121]
Peanuts, red grape, wine	Polyphenol	Increases monoamine and BDNF levels	[121]
Curcumin	Polyphenol	Elevates serotonin, Noradrenaline, and dopamine levels via altering MAO activity	[183,184]
Green tea/ epigallocatechin	Epigallocatechin-3-gallate (EGCG).	Free radical scavenging and antioxidative properties Green tea treatment can reduce HPA axis hyperactivity in response to stress	[185,186,187]
Resveratrol	Polyphenol	Elevates 5-HT and nor epinephrine levels in pre-frontal cortex (PFC) and upregulates BDNF levels	[188,189,190]

**Table 3 cells-11-01362-t003:** Summary of the clinical trials in treating anxiety/depression with gut dysbiosis.

Interventions	Disease Conditions	Phase	Status	Clinical Trials.gov Identifier
Combination of *Bifidobacterium Longum 35624*^®^ *and 1714™* Probiotics	IBS and depression, anxiety	Phase 2	Completed	NCT04422327
Diet modifications	IBS and depression	-	Completed	NCT00788658
Kynurenine pathway metabolites as novel translational biological markers (observational study)	Depression, gut, IBS	-	Unknown	NCT01304355
Integrative treatment model, conventional treatment	Anxiety and depression	-	Completed	NCT01631500
Transcutaneous vagus nerve stimulation	Anxiety and IBS	-	Completed	NCT03440255
Dietary fiber supplementation	Anxiety and IBS	-	Not yet recruiting	NCT04619095
Linaclotide	Anxiety and IBS	-	Not yet recruiting	NCT03342287
Rifaximin	Gut microbiota manipulation, anxiety, and depression	Phase 2	Not yet recruiting	NCT04302402
Specific CBT program (PASCET-PI)	Psychological problems and IBS	-	Unknown	NCT02265588
SAMe (S-adenosyl-L-methionine) and probiotic *Lactobacillus plantarum*	Depression and IBS	-	Completed	NCT03932474
Selegiline	Anxiety, depression, and IBS	Phase 3	Completed	NCT01912391
Multi-strain probiotic product (DSF)	Anxiety, depression, and IBS	-	Not yet recruiting	NCT04006977
Relaxing music	GI abnormalities, anxiety	-	Recruiting	NCT04671628
Information on the microbiome with anxiety	Gut flora in anxiety	No intervention	Recruiting	NCT04211376
Psychotherapy	Depression and IBS	-	Not yet recruiting	NCT04639141
Galacto-oligosaccharides, maltodextrin	Microbiota–gut–brain axis in brain development and mental health	-	Recruiting	NCT03835468
Observational study	Role of gut flora in depression	-	Recruiting	NCT04211467
*Lactobacillus Plantarum* 299v supplementation	Depression, anxiety disorder	Phase 2	Completed	NCT02469545

## Data Availability

The data that support the findings of this study are available in standard research databases such as PubMed, Science Direct, or Google Scholar, and/or on public domains that can be searched with either key words or DOI numbers.

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
