# Peer review of "Mechanistic Insights into the Link between Gut Dysbiosis and Major Depression: An Extensive Review"

_cells, 2022, doi:10.3390/cells11081362_

Round 1
Reviewer 1 Report
The revised manuscript has been greatly improved. I only have a few editorial suggestions. Note: the line numbers are based on the PDF file (on the right side of the page)
- Capitalize each word in the title.
- In the abstract, the second half of the parenthesis is missing after ghrelin (line 30).
- Subtitle 1.3, "objective" should be "objectives".
- Subtitle 2.1, the abbreviation for gut microbiota (GM) has been defined in 1.1, so no need to re-define it at here.
- Subtitle 2.1, line 108, no need to capitalize Gastrointestinal.
- Subtitle 2.1, line 122, the punctuation is missing after "adult [28-30].
- Subtitle 2.1, line 133, account 1% of what? Something is missing in here.
- Subtitle 2.2, line 147, no need to capitalize Gastrointestinal.
- Subtitle 2.2, line 161-163. "In response to chemical and mechanical stimuli, EECs release more than 20 161 different types of signaling molecules. These molecules reach the nucleus tractus solitarius and the hypothalamus in CNS, ..." Do you mean these gut hormones can reach the nucleus tractus solitarius and the hypothalamus? Can they go across the blood-brain barrier at all? Please double check.
- Subtitle 5.2, Table 2 and Table 3 were referred earlier in the text than Table 1. Table 1 was first referenced under subtitle 5.5 (line 613). Usually the table should be numbered according to the order of being referenced in the text.
Author Response
Dear Sir,
Please see the attachment

Reviewer 2 Report
Comments and Suggestions for Authors
This is an extensive and comprehensive review that summarizes the link between gut dysbiosis and depression. The authors also report information on the recent therapies/supplements targeting the gut-brain axis for the effective management of depressive disorders.
There are a few minor comments:
- Page 5, Section 3.1, Line 212: The authors should replace the term “gastrointestinal tract” with the appropriate abbreviation “GI tract”
- Page 6, Figure 1: The figure and its description in the legend are not clear
- Page 8, Figure 2: In the legend, the authors doesn’t mention the decreased tryptophan catatabolism and kynurenine that they show in the figure
- Page 9, Section 3.5, Line 358: The authors should insert the space between microbiota and either
- Page 12, Section 5: The authors discuss the use of therapeutics modulating the gut microbiota in depressive behavior and anxiety, such as probiotics and prebiotics. To complete the update findings regarding the use of recent therapies targeting the gut microbiota I suggest to include a paragraph to discuss the use of postbiotics in depressive disorders.
Author Response
Dear Sir,
Please see the attachment

This manuscript is a resubmission of an earlier submission. The following is a list of the peer review reports and author responses from that submission.
Round 1
Reviewer 1 Report
This paper is an interesting contribution highlighting the possible pathogenetic link between Gut Dysbiosis and Major Depression. For this reason, I consider appropriate the choice to structure the work in paragraphs specifically dedicated to the single aspects of this hypothesized link. However, I believe that the article can be improved, making the discussion of the topic more precise and understanding easier.
Title
I would suggest re-evaluating the choice of the expression "mental depression" in the title.
Section 1. Introduction
I would recommend reviewing the first part of this paragraph (lines 41-50) where depressive disorder is presented superficially, in a confusing and poorly organized manner. It would be advisable to provide an adequate definition of Major Depression and to refer to the fact that we speak exclusively of this type of depression without reference to the depressive phase of Bipolar Disorder. A better presentation of the epidemiological and psychopathological picture of the disorder should therefore be provided. I would also suggest inserting a definition of intestinal dysbiosis (line 51) to begin introducing the concept and to explain the objectives of the review (lines 81-84) by referring to the order in which the different paragraphs appear in the text.
Section 2. Depression and gut microbiota
I believe that this paragraph should be dedicated exclusively to the compositional and functional description of the intestinal microbiota. I would recommend changing the title and deleting the part (lines 94-97) in which the gut microbiota-depression correlation is unclearly referred to in the next paragraph. A definition of microbiome should also be given in this paragraph (line 91).
Section 5. Clinical evidence
Based on the title and content of this paragraph I would suggest deleting the data relating to preclinical studies (lines 211-213). I also believe that the reference to the concept of individual specificity of the microbiome (lines 213-217) is not very coherent with the rest of the paragraph, so I would suggest eliminating it or to motivate and explain the reason for this reference at this level of the discussion. It would be appropriate to further investigate the clinical data that highlight a link between intestinal dysbiosis and Major Depression, so I suggest referring to the review "Gut microbiota and Bipolar Disorder: an overview on a novel biomarker for diagnosis and treatment", as well as further references.
Section 9. Oxidative and nitrosative stress
I would recommend explaining the expression "Pathobionts" (line 368) which has never been used before in order to avoid confusion in understanding the text.
Section 11. Homocysteine
I believe that in the last part of this paragraph (lines 461-464) the link between the ability to produce homocysteine ​​by some pathogenic constituents of the intestinal microbiota and the development of depression is not clear. I would therefore recommend to better explain this type of pathogenetic relationship, providing references to motivate it.
Section12. Strategies targeting gut dysbiosis-associated depression
This concluding paragraph is confusing and poorly organized, despite the importance of the subject matter. It should be reworked and improved so as to explain what the effective therapeutic strategies targeting the intestinal microbiota can be in the treatment of Major Depression. In this sense, it would be advisable to provide precise data on how dietary-nutritional supplements and supplements based on prebiotics and probiotics can improve depressive symptoms (I would recommend referring again to the review "Gut microbiota and Bipolar Disorder: an overview on a novel biomarker for diagnosis and treatment "). I also believe it is necessary to describe fecal microbiota transplantation (FMT) and to explain on the basis of which data we consider a therapeutic potential for resistant depression. However, I would suggest writing a separate paragraph for conclusions that otherwise lose importance.
I recommend that you check the various abbreviations as well as the taxonomic rules for naming the germs making up the microbiota in all part of the text.
Author Response
Comment: This paper is an interesting contribution highlighting the possible pathogenetic link between Gut Dysbiosis and Major Depression. For this reason, I consider appropriate the choice to structure the work in paragraphs specifically dedicated to the single aspects of this hypothesized link. However, I believe that the article can be improved, making the discussion of the topic more precise and understanding easier.
Title
I would suggest re-evaluating the choice of the expression "mental depression" in the title.
Response – Thanks for the comments. The term "mental depression" in the title is changed to “major depression” in the revised manuscript.
Section1. Introduction
I would recommend reviewing the first part of this paragraph (lines 41-50) where depressive disorder is presented superficially, in a confusing and poorly organized manner. It would be advisable to provide an adequate definition of Major Depression and to refer to the fact that we speak exclusively of this type of depression without reference to the depressive phase of Bipolar Disorder. A better presentation of the epidemiological and psychopathological picture of the disorder should therefore be provided. I would also suggest inserting a definition of intestinal dysbiosis (line 51) to begin introducing the concept and to explain the objectives of the review (lines 81-84) by referring to the order in which the different paragraphs appear in the text.
Response – Definition of depression, especially major depressive disorder, is changed appropriately in the revised manuscript. The clinical symptoms, epidemiological data and psychopathological studies are included. Definition of gut dysbiosis is included as per the suggestion, and the objectives of the review are improved and presented in the relevant order in the revised manuscript.
Section 2. Depression and gut microbiota
I believe that this paragraph should be dedicated exclusively to the compositional and functional description of the intestinal microbiota. I would recommend changing the title and deleting the part (lines 94-97) in which the gut microbiota-depression correlation is unclearly referred to in the next paragraph. A definition of microbiome should also be given in this paragraph (line 91).
Response – This section is entirely revised. New headings and subheadings are added according to the manuscript flow. Both the compositional and functional description of the gut microbiota are included. The relationship between gut dysbiosis and depression are discussed under separate heading and definition of gut microbiome is included in the revised manuscript.
Section 5. Clinical evidence
Based on the title and content of this paragraph I would suggest deleting the data relating to preclinical studies (lines 211-213). I also believe that the reference to the concept of individual specificity of the microbiome (lines 213-217) is not very coherent with the rest of the paragraph, so I would suggest eliminating it or to motivate and explain the reason for this reference at this level of the discussion. It would be appropriate to further investigate the clinical data that highlight a link between intestinal dysbiosis and Major Depression, so I suggest referring to the review "Gut microbiota and Bipolar Disorder: an overview on a novel biomarker for diagnosis and treatment", as well as further references.
Response – The content in the lines 211-213 are removed in the revised manuscript. The concept of individual specificity of the microbiome (lines 213-217) are re-located to the section named “composition and dynamics of the gut microbiota”, which is found to be a relevant content. Clinical data linking depression and gut dysbiosis are discussed under separate section named “clinical evidences” in the revised manuscript. The suggested article was referred.
Section 9. Oxidative and nitrosative stress
I would recommend explaining the expression "Pathobionts" (line 368) which has never been used before in order to avoid confusion in understanding the text.
Response – The expression of the term “Pathobionts" is defined under the section named “gut dysbiosis” or “gut microbial dysbiosis”.
Section 11. Homocysteine
I believe that in the last part of this paragraph (lines 461-464) the link between the ability to produce homocysteine ​​by some pathogenic constituents of the intestinal microbiota and the development of depression is not clear. I would therefore recommend to better explain this type of pathogenetic relationship, providing references to motivate it.
Response – The pathogenetic relationship between gut dysbiosis and increased homocysteine production are explained appropriately in the revised manuscript.
Section12. Strategies targeting gut dysbiosis-associated depression
This concluding paragraph is confusing and poorly organized, despite the importance of the subject matter. It should be reworked and improved so as to explain what the effective therapeutic strategies targeting the intestinal microbiota can be in the treatment of Major Depression. In this sense, it would be advisable to provide precise data on how dietary-nutritional supplements and supplements based on prebiotics and probiotics can improve depressive symptoms (I would recommend referring again to the review "Gut microbiota and Bipolar Disorder: an overview on a novel biomarker for diagnosis and treatment "). I also believe it is necessary to describe fecal microbiota transplantation (FMT) and to explain on the basis of which data we consider a therapeutic potential for resistant depression. However, I would suggest writing a separate paragraph for conclusions that otherwise lose importance.
Response – The strategies targeting gut dysbiosis-associated depression is a separate section which discusses about non-therapeutic potential alternative approaches used in the treatment of depression. Data on the trials which have shown improvement in depressive symptoms using dietary-nutritional supplements and supplements like prebiotics, probiotics and synbiotics are included with appropriate references in the revised manuscript. Under separate section, fecal microbiota transplantation (FMT) is explained. Conclusion is reworked entirely in the revised manuscript.
I recommend that you check the various abbreviations as well as the taxonomic rules for naming the germs making up the microbiota in all part of the text.
Response – Abbreviations are made and used correctly in the appropriate places in the revised manuscript. The taxonomic rules for naming the microorganisms are revised and italicized correctly in the revised manuscript.
"Please see the attachment"

Reviewer 2 Report
This reviewer would like to thank the authors for their efforts to carry out this extensive review. Depression and mood disorders affect a large part of the people and its interconnection with the gut brain axis is very interesting. The complexity of this association is overwhelming and collecting clinical and preclinical studies is complicated. According to my opinión, the authors do not manage to make the reading of this revision profitable, despite being well referenced. First of all it seems that each section is written by a different author, giving the revision little consistency. I recommend to the authors a greater concreteness, that is, that they do not repeat aspects commented in previous sections. I recommend to the authors a better structure of the sections, a better use of the "point and apart", since the reading becomes very complicated if everything is written from one unique paragraph. the structure of the sections sometimes leads to confusion, the results of the referenced studies are not clearly exposed. There are times when it is not known if it is a clinical study or an animal model study. I suggest authors do a thorough exercise of looking at the review as a whole for greater compression and appropriate reading. Because of all this I recommend to the editors that this work be subjected to a major revision taking into account what was said above.
With respect to minor revisions abbreviations should be consistent throughout the work. The first one that is named is put the complete name (abbreviation) and the rest of the times the abbreviation is put. Review the paper and you will see that this is not always the case.
The name of the sections can be improved in many cases and must provide more information about what is going to be exposed.
Only bacterial genera must be written itálica. Families, phyla, not in italics.
Author Response
Comments: This reviewer would like to thank the authors for their efforts to carry out this extensive review. Depression and mood disorders affect a large part of people and its interconnection with the gut-brain axis is very interesting. The complexity of this association is overwhelming and collecting clinical and preclinical studies is complicated. According to my opinion, the authors do not manage to make the reading of this revision profitable, despite being well referenced. First of all, it seems that each section is written by a different author, giving the revision little consistency. I recommend to the authors a greater concreteness, that is, that they do not repeat aspects commented in previous sections. I recommend to the authors a better structure of the sections, better use of the "point and apart" since the reading becomes very complicated if everything is written from one unique paragraph. The structure of the sections sometimes leads to confusion, the results of the referenced studies are not clearly exposed. There are times when it is not known if it is a clinical study or an animal model study. I suggest authors do a thorough exercise of looking at the review as a whole for greater compression and appropriate reading. Because of all this I recommend to the editors that this work be subjected to a major revision taking into account what was said above.
Response – All the authors appreciated the reviewer’s time and effort in reviewing our manuscript. We also appreciate the reviewers’ expert comments to improve our manuscript. As per the comments, the structure of the entire manuscript, the headings and subheadings of the sections, and their respective order is revised thoroughly. Relevant contexts under each heading and subheadings are added. The contexts are abstracted and the repetition of the information is removed. Comprehension, content flow, and consistency are appropriately revised and checked. In each example of the research studies, the respective animal model study or clinical study is highlighted. The headings depict the information in a coherent order to improve the reading.
- With respect to minor revisions, abbreviations should be consistent throughout the work. The first one that is named is put the complete name (abbreviation) and the rest of the times the abbreviation is put. Review the paper and you will see that this is not always the case.
Response – We appreciate the comment, which we fully agree with. Abbreviations are now corrected and consistent throughout the work in the revised manuscript.
- The name of the sections can be improved in many cases and must provide more information about what is going to be exposed.
Response –Headings and sub-heading of the sections are relevantly changed in the revised manuscript, according to the content included under each section.
Only bacterial genera must be written itálica. Families, phyla, not in italics.
Response – Appropriate italicization of the microorganisms’ names is made in the revised manuscript.

Reviewer 3 Report
This manuscript is an extensive review on the possible connections between gut dysbiosis and depression. It would provide significant contribution to the field. However, I do have a few comments that may help to clarify this paper.
- In abstract, “The etiology of depression is often multifactorious with sex, genetic, environmental, psychological, etc.” A noun is missing after the adjectives (sex, genetic, environmental, psychological).
- In abstract, “Gut microbiome communicates with the brain through the neural, immune, and metabolic pathways either directly (via vagal nerves) or indirectly.” Hormones are also involved and should be included in this list.
- The introduction is too long. It can be consolidated to one page or a bit over a page.
- In introduction, “Even though, a variety of factors such as genetic, environmental, psychological, and biochemical account for the etiology of depression, the common age range for the onset usually falls within 15-30 years of age.” The two halves of the sentence do not have a contrast relationship and might be split into two independent sentences.
- In introduction (1.1), “Recent studies in human microbial ecosystem have shown the important physiological role of gut microbiota in maintaining the gastrointestinal (GI), hormonal, immune and neural homeostasis, called as gut-brain axis (GBA).” This definition of gut-brain axis is different from the definition that is given later in the manuscript (page 5, lines 10-12), which is stated as “Brain and gut microbiome communicate bidirectionally via major pathways like vagus nerve, neuroendocrine, neuroimmune, autonomic nervous system (ANS), ENS, and the humoral links, otherwise termed as GBA.” Please be consistent with your definition of the gut-brain axis.
- In introduction (1.1), “The neural transmission was reported to be disrupted by GD associated with gut leakiness and local inflammation [18].” Please specify where specifically the neural transmission was disrupted by gut dysbiosis. Does it happen in the brain or the enteric nervous system?
- Page 3, 2.1. The composition, dynamics and functions of GM. “Gut microbiome is composed of more than 1,014 microorganisms, … … Microbiota comprises approximately 1,100 predominant and 160 other species types.” Do you mean 1,014 types of microorganisms? Why is this number different from the following sentence?
- Page 4, 2.2. Microbial-derived metabolites and their physiological functions. “These molecules reach the digestive behavior centers in the CNS (such as the nucleus tractus solitarius and the hypothalamus) and activate the afferent vagal terminals to generate the GBA signals [42].” This sentence is confusing. Would activating the afferent vagal terminals happen first and reaching to the nucleus tractus solitarius and hypothalamus happen after? The sequence of events does not sound correct.
- Page 4, 2.2. Microbial-derived metabolites and their physiological functions. “Bile acids activate the nuclear receptor farnesoid X in the ileum, resulting in the generation of fibroblast growth factor 19 or its ortholog fibro-blast growth factor 15, which reaches the systemic circulation and penetrates the blood-brain barrier (BBB) in mice [46–48].” What is the consequence of that? Would that cause any depression like symptoms?
- Page 5, 3. Gut microbial dysbiosis or gut dysbiosis. If “gut microbial dysbiosis” and “gut dysbiosis” mean the same thing. Would it be OK to use just one term to keep it simple?
- Subtitle 3 talks some pathological mechanisms underlying gut dysbiosis-induced depressive symptoms, which is redundant to the next subtitle. Thus, it might be better to combine subtitle 3 and subtitle 4.
- Page 6, 4.1 GBA dysregulation in depression. “GD-induced shifts in SCFA composition reduce the 5-HT levels”. Where do the 5-HT levels change? In the brain or in the gut?
- Figure 1. (1) There are two ways the HPA axis is illustrated, the green arrowed curve and the flow chart. Please keep only one. (2) Cortisol should be released to the blood, not to the gut lumen. Thus, cortisol should be illustrated to the basolateral side of the intestinal epithelium. (3) “SCF” should be “SCFAs”, to be consistent with the text. (4) Why do you show enterochromaffin cells on this figure? Enterochromaffin cells mainly release histamine, but histamine function in gut-brain axis or depression was not discussed at all. (5) In the figure legend, delete the abbreviations for TJ and AJ because they are not used on the figure. (6) Define LPS.
- Should Figure 3 be Figure 2 and vice versa? Besides, Figure 3 was not referenced in the main text.
- Figure 2. There is not description for (d) and (e).
- Pay attention to punctuations and spaces. There are many places in the document that a space is missing between two adjacent words. Punctuations are also missing in some places.
- Too many abbreviations are used, which slows down reading. Try to reduce the use of abbreviations.
Round 2
Reviewer 2 Report
Despite the efforts of the authors and the length of this review, I will advise you to reject the article so that it can be completely redone in the future if the authors so wish.
Reading becomes very complicated because you go from one topic to another within each section and sometimes it does not coincide with what should be developed in the section. This must be completely reviewed and must be adjusted to the specific topic that they are going to develop. This happens constantly throughout the entire review.
The text is full of errors and requires a more thorough revision of the language, as well as a more fluid writing for the reader.
There is still a lot of confusion when reading the paper about whether we are talking about clinical or preclinical studies.
Most of the knowledge of the gut brain microbiota axis we know from studies in mice, especially in GF mice, I think this should be further developed, highlighting the relevant findings and connecting them with the mechanisms of action. It is a complex topic, but the way it is developed in the paper makes it impossible to take advantage of this review.
There is still redundant information as well as interesting topics to be developed where it is not deepened, such as clinical trials with probiotics in humans, there are even meta-analyses of these topics.
Figure 2 does not have the appropriate resolution, in the tables of the studies it is convenient to put the design of the study and the sample size.
Again there are many names of bacteria that should be written in italics.